# OpenReview forum: "PipelineRL: Faster On-policy Reinforcement Learning for Long Sequence Generation"
_TMLR — Accepted by TMLR_

### Review · Reviewer_pNXA · 2025-11-08

**Summary Of Contributions:**

## Summary of Contributions

This paper investigates the limitation of conventional RL when scaling up, due to insufficient hardware utilization. The authors present PipelineRL, which optimizes the generation-training pipeline from two aspects: (1) raining and generation run in parallel and asynchronously (2) update generation weights in-flight. Through Eq.(7), they analyze both conventional RL and PipelineRL from effectiveness and throughput. Experiments verify the advantage of the proposed method.

## Strength

The paper is easy to follow. From the intuition level, the analysis in Section 3 and 4 make sense to me. The experiments verify the effectiveness of PipelineRL.

## Weakness

1. PipelineRL seems not a quite novel idea. At least to my knowledge, such a framework (or similar ones) has been widely applied in industry. There are also a few similar techniques in published literature. For example, "partial rollout" seems a very relevant technique. In partial rollout, once a training batch is collected, the unfinished queries are paused and only resumed after the weights update is finished.

2. The analysis through effectiveness and throughput is interesting, but I expect it to be more rigorous. Throughout the paper, I did not find a formal statement to quantify $\Delta R/\Delta S$ or $\Delta S/ \Delta t$ (maybe by B,G,N, etc). To explain the advantage of PipelineRL, it would be better to formally show it has both higher effectiveness and throughput than conventional RL.
Fig3-(b) is just a schematic illustration, and it is totally unclear how well the curve and points in the figure aligns with the practice. It can cause some confusion.

3. By Fig5-(b), it seems that Pipeline RL does not improve or decrease the sample efficiency, which seems a bit counter-intuitive. Pipeline RL and conventional RL should have different level of "off-policiness", which is the main factor to determine the sample efficiency. Although I'm not sure which one of them are more off-policy, there should be some divergence in the curve. It would be great if the authors can explain it.

**Audience:**

Yes

**Audience Explanation:**

The focus of the paper is to optimize the RL training process in LLM. LLM post-training is a popular topic, not only for people working on LLM but also for people more on the RL side.

Though, I feel the contribution of this paper is a bit off-topic to the machine learning community, because the main improvement of the proposed PipelineRL, compared with conventional RL, is more on the implementation side rather than innovation on the algorithmic level.

**Broader Impact Concerns:**

I do not have concerns on the ethical implications.

**Claims And Evidence:**

Yes

**Claims Explanation:**

Fig 3-(b) is not plotted with real data but just a schematic illustration. I think it is not very rigorous.

Comparing different methods from the perspective of effectivenss and throughput is interesting, but there is no formal quantification for those quantities. This makes their arguments not very convincing.

**Requested Changes:**

My proposed adjustments are based on my concerns in weakness.

1. I encourage the authors to do a deeper survey on existing relevant techniques and compare with them in related works. For example, I think partial rollout is relevant, but you can correct me if I'm wrong.

2. Provide a formal quantification for $\Delta R/\Delta S$ or $\Delta S/ \Delta t$. Or formally prove and state the proposed PipelineRL has higher effectiveness or throughput than conventional RL (at least in some cases). Remove Fig 3-(b) or replace it with figure plotted with real data.

3. Add explanation for why curves in Fig 5-(b) do not diverge.

1 & 2 are critical to securing my recommendation, and 3 is not very important.

---

> ### Author Response · Authors · 2025-12-30
> **Response to Reviewer pNXA**
>
> We thank the reviewer for their review. We address all their requested changes.
>
> > I encourage the authors to do a deeper survey on existing relevant techniques and compare with them in related works. For example, I think partial rollout is relevant, but you can correct me if I'm wrong.
>
> Partial rollouts usually refers to training on partially completed rollouts using a value function. While we do not discuss partial rollouts explicitly, we cited IMPALA, SEED RL, and OpenAI Five which are asynchronous and learn from partial rollouts. PipelineRL is also asynchronous, but trains on completed rollouts. We added a sentence to highlight this distinction in the related work.
>
> > Provide a formal quantification for $\Delta R/\Delta S$ or $\Delta S/ \Delta t$.
>
> | Method | $\Delta R / \Delta S$ | $\Delta S / \Delta t$ | $\Delta R / \Delta t$ |
> |--------|----------------------|----------------------|----------------------|
> | Conv RL G=8 | $\approx 4.2\times10^{-7}$ | $\approx 55.6$ | $\approx 2.3\times10^{-5}$ |
> | Conv RL G=16 | $\approx 4.2\times10^{-7}$ | $\approx 74.1$ | $\approx 3.1\times10^{-5}$ |
> | Conv RL G=32 | $\approx 4.2\times10^{-7}$ | $\approx 114.9$ | $\approx 4.8\times10^{-5}$ |
> | Conv RL G=64 | $\approx 2.4\times10^{-7}$ | $\approx 165.0$ | $\approx 4.0\times10^{-5}$ |
> | PipelineRL | $\approx 4.2\times10^{-7}$ | $\approx 222.2$ | $\approx 9.3\times10^{-5}$ |
>
> We can clearly see the advantages of PipelineRL in terms of $\Delta R / \Delta t$
>
> > Formally prove and state the proposed PipelineRL has higher effectiveness or throughput than conventional RL (at least in some cases).
>
> A formal proof of PipelineRL higher effectiveness for a given amount of lag is provided in A4.
>
> > Remove Fig 3-(b) or replace it with figure plotted with real data.
>
> We removed Fig 3b from the paper.
>
> > Add explanation for why curves in Fig 5-(b) do not diverge.
>
> Diverging curves were not plotted in Fig 5b because it made reading difficult. Please see A5 for a Conventional RL G=64 which diverges.

---

> > ### Comment · Reviewer_pNXA · 2026-01-02
> >
> > Thanks for the authors' efforts in improving the paper. My main concerns are addressed.

---

### Review · Reviewer_oihW · 2025-11-13

**Summary Of Contributions:**

This paper addresses the significant challenge of effectively scaling RL methods used to enhance the reasoning capabilities of LLMs. The core contribution is PipelineRL, an approach designed to achieve a superior trade-off between hardware efficiency (accelerator utilization) and the freshness of training data (on-policy).

**Audience:**

Yes

**Audience Explanation:**

Scaling up RL training for LLM is an important topic, and many RL people would be interested in knowing the findings.

**Broader Impact Concerns:**

There isn't a significant safety risk beyond the risk of doing RL/AI research and how it might engender our society. No such statement is necessary for this work.

**Claims And Evidence:**

Yes

**Claims Explanation:**

The contribution is largely about system design rather than novelty. This type of asynchronous RL design has been explored in Deep RL (Distributed Asynchronous Actor-Critic A3C [1]) by DeepMind around 2019. However, verifying and validating that the same principle works, and releasing the system design blueprint **publicly** is a good contribution, and can help speed up the effort in the open-source community.

Also, it's worth noting that RL advancements are often closely tied to advances in system design. Even though the paper focuses on system design, it remains of great interest to the RL community.

[1] Horgan, D., Quan, J., Budden, D., Barth-Maron, G., Hessel, M., Van Hasselt, H., & Silver, D. (2018). Distributed prioritized experience replay. arXiv preprint arXiv:1803.00933.

**Requested Changes:**

Some of my questions are about making this paper more scientific instead of pure engineering:
- Not enough analysis/results were around the trade-off between synchronization frequency (in-flight weights update) and final policy performance -- how should we set these hyperparameters? If I missed such an analysis, please refer to the sections in the paper or plots.
- Figure 3 is very interesting in highlighting the difference between Pipeline RL scaling and synchronized RL scaling. However, there isn't any empirical exploration on how the scaling happens. Can you comment on how Pipeline RL can scale up?
- It's a little unclear what a good future direction should be. You don't have to do them, but any ideas on what other researchers can explore?

---

> ### Author Response · Authors · 2025-12-30
> **Response to Reviewer oihW**
>
> We thank the reviewer for their review. We address all their requested changes.
>
> > Not enough analysis/results were around the trade-off between synchronization frequency (in-flight weights update) and final policy performance -- how should we set these hyperparameters? If I missed such an analysis, please refer to the sections in the paper or plots.
>
> In PipelineRL, in-flight weight updates happen after every optimizer step. Reducing the synchronization frequency would increase the off-policyness. However, we study in details the effects of increasing the number of optimizer step G per RL step in the context of Conventional RL.
>
> > Figure 3 is very interesting in highlighting the difference between Pipeline RL scaling and synchronized RL scaling. However, there isn't any empirical exploration on how the scaling happens. Can you comment on how Pipeline RL can scale up?
>
> We scaled PipelineRL up to 16 nodes without issues. At very large scale, increasing off-policyness could potentially result in unstable learning, which might be addressed by reducing the learning rate or increasing the batch size. We leave the systematic exploration of these hyper-parameters change to future work, as our current 16-node experiments remained stable throughout training.
>
> > It's a little unclear what a good future direction should be. You don't have to do them, but any ideas on what other researchers can explore?
>
> We outlined future directions in Section 7:
>
> 1. Applying PipelineRL to agentic behaviors with interleaved LLM generations and environment interactions. The variance between trajectories can be very high since some of them might multiple turns and some single turns.
> 2. Studying the differential impact of token lag across sequence positions (when do low-lag recent tokens help vs. high-lag early tokens hurt?)
> We also added this new direction:
> 3. There might exist better RL algorithms when combined with PipelineRL than traditional algorithms such as REINFORCE with truncated importance sampling.

---

### Review · Reviewer_snwq · 2025-12-09

**Summary Of Contributions:**

This work introduces PipelineRL, a novel training pipeline designed to speed up on-policy reinforcement learning for LLMs, especially in long sequence generation tasks. Specifically, the key idea is to concurrently generate data and train the model in an asynchronous manner, enabled by a novel "in-flight" weight update mechanism. That is, the generation process only pauses briefly to receive updated model weights and then continues generating tokens, which maximizes GPU utilization while keeping the training data nearly on-policy. In addition, the authors also provide empirical demonstration of substantial speedup. Particularly, on a mathematical reasoning benchmark using a 7B LLM and deployed over 128 GPUs, the method achieves roughly 2 $\times$ faster learning to reach comparable performance relative to a standard two-stage RL baseline. Moreover, the authors show that despite asynchronous updates, the data remains sufficiently on-policy and final model performance on benchmarks remains competitive.

**Audience:**

Yes

**Audience Explanation:**

There is a growing interest in training LLMs with RL (e.g., RLHF and other goal-driven fine-tuning of LLMs), as well as a general interest in techniques for scaling up machine learning algorithms efficiently. This work sits at the intersection of those areas by addressing how to make on-policy RL feasible and faster on cutting-edge hardware for very long sequence generation. Researchers and practitioners working on large-scale RL, distributed training, or LLM optimization will likely be interested in this work.

**Broader Impact Concerns:**

The work does not raise any obvious ethical or societal concerns. It is focused on a training methodology to improve efficiency, with no new application or data that might be sensitive.

**Claims And Evidence:**

Yes

**Claims Explanation:**

The authors provide  empirical evidence to support their main claims. Regarding the efficiency, PipelineRL consistently learns faster than the baseline. For example, on a long-form mathematical reasoning benchmark, PipelineRL reaches a given reward level roughly twice as fast as a standard synchronous RL setup, which is shown in their Figure 5.
Additionally, regarding the on-policyness and training stability of PipelineRL, they measure metrics like the effective sample size and KL divergence to quantify how "on-policy" the data remains. The results show that despite the asynchronous updates, which introduce some lag in using slightly older weights for part of a sequence, PipelineRL's data distribution stays nearly as on-policy as a tightly synchronous RL with minimal lag, which is supported by Figure 6.

**Requested Changes:**

1.  The experiments of this paper focus on a single task domain (math reasoning task) and a specific model architecture and scale. Including another type of long-sequence generation (e.g., long-form text generation, dialogue, or code generation) and other choices of model would strengthen generality claims.

2. The paper introduces the risk of stale key-value caches when weights are updated during sequence generation and notes that recomputing the cache can mitigate this at some cost. It would help to add a bit more detail on how this was handled in practice. For example, did the authors find it necessary to recompute the cache periodically, or did training work well without any recomputation? Was there any observable impact of stale caches on the generated tokens or rewards?

3. Minor point: Caption in Figure 7: "checkpoint" ---> "checkpoints"

---

> ### Author Response · Authors · 2025-12-30
> **Response to Reviewer snwq**
>
> We thank the reviewer for their review. We address their requested changes.
>
> > The experiments of this paper focus on a single task domain (math reasoning task) and a specific model architecture and scale. Including another type of long-sequence generation (e.g., long-form text generation, dialogue, or code generation) and other choices of model would strengthen generality claims.
>
> Training a model from scratch to improve mathematical reasoning is particularly difficult since the policy goes through quick changes during training and sequence lengths grow dramatically (as noted in Guo et al., 2025). These characteristics stress-test PipelineRL's ability to maintain on-policyness.
>
> The core architectural contributions, concurrent asynchronous generation/training and in-flight weight updates, are not domain dependent. The improvement will translate to other domains also having variability in sequence lengths.
>
> > The paper introduces the risk of stale key-value caches when weights are updated during sequence generation and notes that recomputing the cache can mitigate this at some cost. It would help to add a bit more detail on how this was handled in practice. For example, did the authors find it necessary to recompute the cache periodically, or did training work well without any recomputation? Was there any observable impact of stale caches on the generated tokens or rewards?
>
> We did not need to recompute the KV cache. As shown in Section 5.1 and Figure 7, we empirically evaluated the impact of stale KV caches by comparing PipelineRL with and without KV cache recomputation across three training checkpoints. Our results demonstrate that using stale KV cache introduces only minimally higher KL divergence compared to recomputing it (see Figure 7). We have added a sentence in the experimental setup paragraph.
>
> > Minor point: Caption in Figure 7: "checkpoint" ---> "checkpoints"
>
> Fixed.

---

### Author Response · Authors · 2025-12-30
**General response**

We addressed every requested changes and concerns and updated the pdf accordingly. The changes in the pdf are in green to make it easier to review.

---

### Decision · Action_Editor_KuUe · 2026-01-30

**Recommendation:** Accept as is

**Audience:**

Yes

**Audience Explanation:**

LLMs and their training are relevant to parts of TMLR's audience. Hence scaling up RL training for LLMs is as well. Also some people working in the RL space might be interested in the findings.

**Claims And Evidence:**

Yes

**Claims Explanation:**

The experimental evidence supports the paper's claims. Several experiments and results are presented that also shed lights on aspects of the proposed approach regarding scalability and efficiency.

While broader validation across additional tasks and models would strengthen generality, the presented results are sufficient for acceptance given the work’s practical relevance.